

# Interhemispheric transport of metallic ions within ionospheric sporadic E layers by the lower thermospheric meridional circulation

Bingkun Yu[1,2], Xianghui Xue[1,3,4,5], Christopher J. Scott[2], Jianfei Wu[1,3,4], Xinan Yue[6], Wuhu Feng[7,8], Yutian Chi[1,2], Daniel R. Marsh[7,9], Hanli Liu[10], Xiankang Dou[1,5,11], and John M. C. Plane[7]

[1]CAS Key Laboratory of Geospace Environment, Department of Geophysics and Planetary Sciences, University of Science and Technology of China, Hefei, China
[2]Department of Meteorology, University of Reading, Berkshire, UK
[3]CAS Center for Excellence in Comparative Planetology, Hefei, China
[4]Anhui Mengcheng Geophysics National Observation and Research Station, University of Science and Technology of China, Hefei, China
[5]Hefei National Laboratory for the Physical Sciences at the Microscale, University of Science and Technology of China, Hefei, China
[6]Key Laboratory of Earth and Planetary Physics, Institute of Geology and Geophysics, Chinese Academy of Sciences, Beijing, China
[7]School of Chemistry, University of Leeds, Leeds, UK
[8]National Center for Atmospheric Science, University of Leeds, Leeds, UK
[9]National Center for Atmospheric Research, Boulder, CO, USA
[10]High Altitude Observatory, National Center for Atmospheric Research, Boulder, CO, USA
[11]Wuhan University, Wuhan, China

**Correspondence:** Xianghui Xue (xuexh@ustc.edu.cn); Christopher J. Scott (chris.scott@reading.ac.uk)

**Abstract.** Long-lived metallic ions in the Earth's atmosphere/ionosphere have been investigated for many decades. Although the seasonal variation in ionospheric 'sporadic E' layers was first observed in the 1960s, the mechanism driving the variation remains a long-standing mystery. Here we report a study of ionospheric irregularities using scintillation data from COSMIC satellites and identify a large-scale horizontal transport of long-lived metallic ions, combined with the simulations of the Whole

5  Atmosphere Community Climate Model with the chemistry of metals and ground-based observations from two meridional chains of stations from 1975–2016. We find that the lower thermospheric meridional circulation influences the meridional transport and seasonal variations of metallic ions within sporadic E layers. The winter-to-summer, meridional velocity of ions is estimated to vary between -1.08 and 7.45 m/s at altitudes of 107–118 km between 10°–60°N latitude. Our results not only provide strong support for the lower thermospheric meridional circulation predicted by a whole atmosphere chemistry-climate

10  model, but also emphasise the influences of this winter-to-summer circulation on the large-scale interhemispheric transport of composition in the thermosphere/ionosphere.

## 1  Introduction

Sporadic $E$ ($E_s$) layers of metallic ion plasma occur in the E region at altitudes between 90 and 130 km. Unlike other ionospheric layers, formed by the photoionization of $N_2$ and $O_2$, in which the major molecular ions $NO^+$ and $O_2^+$ have lifetimes of



order seconds, the $E_s$ layer is composed of long-lived (up to 10 days) metallic ions (Plane et al., 2015) and is remarkably thin, typically 1–3 km thick (Layzer, 1972). Over one third of the interruptions to the Global Navigation Satellite System caused by ionospheric weather can be attributed to the $E_s$ layer (Yue et al., 2016). One long-standing mystery is the marked seasonal variability of $E_s$, with a maximum between 10°–60° latitude in the summer hemisphere and a minimum between 60°–70°

latitude in the winter hemisphere (Whitehead, 1989; Tsai et al., 2018; Yu et al., 2019b). The most likely explanation for the seasonal dependence of Es layers is the wind shear, by which the seasonal variations of zonal and meridional winds in the E region above 95 km make the summer E region dominated by the vertical convergence of ions, but dominated by the diffusion of ions in winter (Yuan et al., 2014). Model-simulated mean divergences of the concentration flux of metallic ions by vertical wind shear are mainly distributed in the mid-latitudes of 20°–40° in winter (Wu et al., 2005; Chu et al., 2014; Yu et al., 2019b,

2020). A fundamental disagreement of the distribution of the $E_s$ winter minimum exists between observations and recent model simulation, particularly when the effects of magnetic declination angle were considered (Yu et al., 2019b). Although $E_s$ layers have been investigated since the early years of ionospheric investigations (Whitehead, 1960), their seasonal dependence cannot be explained by the vertical wind shear theory (Whitehead, 1989). This remains one of the weakest points in understanding $E_s$ layer formation. Recently, the sporadic E-like phenomena were discovered in the Martian ionosphere (Collinson et al., 2020),

which indicates that understanding $E_s$ layers and the dynamical/electrodynamical processes that perturb planetary ionospheres is very important for long-distance radio communication on Earth, Mars and other planets.

    Numerical studies indicate that there is a winter-to-summer meridional circulation in the lower thermosphere driven by gravity-wave forcing. This meridional circulation is referred to as the lower thermospheric residual mean meridional circulation (Liu, 2007). Recent modeling studies (Smith et al., 2011; Qian et al., 2017; Qian and Yue, 2017) have found clear signatures

that the residual meridional circulation is important for trace gas distributions and composition variations in the thermosphere and ionosphere. However, the lower thermospheric meridional wind is considerably smaller than the amplitudes of tides (Wu et al., 2008), so that it is difficult to directly measure and characterize.

    The present paper reports a study of the global-scale winter-to-summer transport of metallic ions in the upper atmosphere, from scintillation data observed by the Constellation Observing System for Meteorology, Ionosphere, and Climate (COSMIC)

satellites (Anthes et al., 2008). The windshear theory is the most likely candidate for explaining the interhemispheric transport of metallic ions within Es layers. We extend the vertical windshear theory (Mathews, 1998) to three dimensions, deriving the zonal, meridional and vertical ion velocities. The large-scale horizontal transport of ions is analysed, with respect to the mechanism of $E_s$ formation and a possible impact of the winter-to-summer lower thermospheric circulation on the morphology of $E_s$ layers. The velocity of the meridional transport of metallic ions is quantitatively estimated, from two meridional chains

of ground-based ionospheric monitoring stations, located roughly along the Greenwich meridian (0°E) and 120°E longitude. These ionospheric observations cover a period of 35 years. The quantitative assessment is consistent with the meridional ion velocity predicted by model calculations.





## 2 Data and Method

The COSMIC multi-satellite dataset was used to investigate the global morphology of $E_s$ layers during the period 2006-2011. For this we used the COSMIC-GPS amplitude scintillation index S4 (Yu et al., 2020), including the maximum S4 index (S4max), and geographic latitude, longitude, altitude and time at which S4max was detected. The amplitude of the S4 index is

one of the most important parameters in the scintillation data, defined as the standard deviation of the normalized intensity in signals. Large S4 values indicate ionospheric plasma irregularities with strong fluctuations (Yue et al., 2015; Tsai et al., 2018), which are related to the plasma frequencies of Es layers (Arras and Wickert, 2018; Resende Chagas et al., 2018). The Es layer is a thin layer of enhanced plasma irregularities and is composed of metallic ions which converge vertically mainly due to wind shear (Kopp, 1997; Mathews, 1998; Grebowsky and Aiken, 2002). The Es layer scatters, refracts, or reflects incident HF/VHF

radio waves (Whitehead, 1989). It has been demonstrated that there is a universal connection between S4max occurring at heights of 90–130 km and critical frequencies of Es layers (Yu et al., 2020). The S4max occurring at $E_s$ altitudes of 90–130 km are used as a proxy for the electron concentration within $E_s$ layers (Wu et al., 2005; Yue et al., 2015; Arras and Wickert, 2018; Resende Chagas et al., 2018; Yu et al., 2019b).

The long-term ground-based observations of intensities of $E_s$ layers represented by the critical frequencies $f_oEs$ from two

meridional chains of low-to-middle latitude ionosonde stations were used in this study. These are five ionospheric sounders (ionosondes) located roughly along the Greenwich meridian ($0°$E) at Lerwick ($60.13°N, 1.18°W$), Slough ($51.51°N, 0.60°W$), Poitiers ($46.57°N, 0.35°E$), Lisbon ($38.72°N, 9.27°W$), Ouagadougou ($12.37°N, 1.53°W$) between 1975 and 1998; and five digital ionosondes (digisondes) (Bibl and Reinisch, 1978) located roughly along $120°$E longitude at Mohe ($52.0°N, 122.5°E$), Beijing ($40.3°N, 116.2°E$), Wuhan ($30.5°N, 114.4°E$), Shaoyang ($27.1°N, 111.3°E$), Sanya ($18.3°N, 109.4°E$) between

2006 and 2016 under the Chinese Meridian Project (Wang, 2010).

The lower thermospheric residual circulation is calculated using the Whole Atmosphere Community Climate Model (WACCM). WACCM is a global climate model with interactive chemistry, developed at the National Center for Atmospheric Research. A specified dynamics version of WACCM4 (SD-WACCM4) (Marsh et al., 2013) is used. SD-WACCM4 is constrained by relaxing the temperature and horizontal winds to those from the Modern-Era Retrospective Analysis for Research and Appli-

cations, which is a NASA reanalysis for the satellite era by a major new version of the Goddard Earth Observing System Data Assimilation System Version 5 (Rienecker et al., 2011).

It is widely accepted that the mechanism for $E_s$ formation at mid-latitudes is wind shear. From the vertical windshear theory (Mathews, 1998), the vertical motion of metallic ions by the neutral winds is described by equation (1):

$$w_i = \frac{r_i cosI}{1+r_i^2} \times U - \frac{sinIcosI}{1+r_i^2} \times V + \frac{r_i^2 + sin^2I}{1+r_i^2} \times W \qquad (1)$$

where $w_i$ represents the vertical velocity of ions, $I$ represents the dip angle of the geomagnetic field $\boldsymbol{B}$ (positive in the downward direction in the Northern Hemisphere), $r_i$ is the ratio of the ion-neutral collision frequency ($\nu_i$) to the ion gyrofrequency ($\omega_i$), and $\boldsymbol{V}_n$ $(U, V, W)$ represents the neutral wind in zonal (positive for eastward), meridional (positive for northward), and vertical (positive for upward).



From Eq. (1) which describes the classical windshear theory, the zonal and meridional winds act upon the vertical plasma convergence of metallic ions at all altitudes, as weighted by the ratio $r_i$ of $\nu_i$ to $\omega_i$. However, the meridional component of ion velocity should also be included in the windshear theory. The formation of an $E_s$ layer is controlled fully by the ion dynamics, and expressed through a basic ion momentum equation (Chimonas and Axford, 1968) (equation 2 below). The steady-state ion momentum equation includes the ion-neutral collision and geomagnetic Lorentz forces:

$$m_i \frac{\mathrm{d}\boldsymbol{v}_i}{\mathrm{d}t} = 0 = e(\boldsymbol{E} + \boldsymbol{v}_i \times \boldsymbol{B}) - m_i \nu_i (\boldsymbol{v}_i - \boldsymbol{V}_n) \tag{2}$$

where $m_i$ is the ion mass, $\boldsymbol{B}(BsinDcosI, BcosDcosI, -BsinI)$ is the magnetic field, $I$ and $D$ are the dip angle and declination angle of the magnetic field, $\boldsymbol{v}_i$ $(u_i, v_i, w_i)$ is the ion drift velocity, and $\boldsymbol{V}_n$ $(U, V, W)$ is the neutral wind. We derive the ion drift in the zonal, meridional and vertical directions from Eq. (2), by adopting an east, north and vertically upward Cartesian coordinate system. When the electric filed $\boldsymbol{E}$ is neglected, a conventional presentation of the wind shear theory is derived. As the vertical windshear is described in the vertical component, we refer to this extended view of the formation and transport of the layering process as "generalized windshear theory" with three dimensional motions of ion drift:

$$u_i = \frac{r_i^2 + sin^2 D cos^2 I}{1 + r_i^2} U + \frac{-r_i sinI + sinD cosD cos^2 I}{1 + r_i^2} V - \frac{r_i cosD cosI + sinD sinI cosI}{1 + r_i^2} W \tag{3}$$

$$v_i = \frac{r_i sinI + sinD cosD cos^2 I}{1 + r_i^2} U + \frac{r_i^2 + cos^2 D cos^2 I}{1 + r_i^2} V + \frac{r_i sinD cosI - cosD sinI cosI}{1 + r_i^2} W \tag{4}$$

$$w_i = \frac{r_i cosD cosI - sinD sinI cosI}{1 + r_i^2} U - \frac{r_i sinD cosI + cosD sinI cosI}{1 + r_i^2} V + \frac{r_i^2 + sin^2 I}{1 + r_i^2} W \tag{5}$$

## 3 Results

The maps in Figures 1a and 1b show the geographical distribution of $E_s$ layers represented by the S4max index for the winter (December, January, February) and summer (June, July, August), based on the six-year COSMIC multi-satellite dataset between 2006 and 2011. The most dramatic feature of the global $E_s$ map is the pronounced summer maximum in a broad band of 10°–60°. $E_s$ layers are weaker in winter. The winter minimum of $E_s$ (S4max<0.3) is observed along the 60°–80° geomagnetic latitude band, dividing the mid-latitude region and polar region. This typical $E_s$ seasonal dependence has been revealed in both hemispheres over the past decades, from ground-based ionosondes, radar and satellite measurements (Whitehead, 1989; Mathews, 1998; Wu et al., 2005; Haldoupis et al., 2007; Arras et al., 2008; Chu et al., 2014; Yu et al., 2019b, 2020).

Figures 1c and 1d show the atmospheric circulation in winter and summer averaged from 2006–2011, calculated using SD-WACCM4. Three circulation cells are apparent, which are consistent with the previous studies (Qian et al., 2017; Qian





and Yue, 2017). Below ∼95 km, the summer-to-winter mesospheric circulation is driven by gravity-wave forcing. Between ∼95 km and ∼115 km, the winter-to-summer lower thermospheric circulation is driven by gravity-wave forcing that is in the opposite direction to that which drives the mesospheric circulation. Above ∼115 km, the thermospheric circulation is a summer-to-winter circulation driven by solar EUV heating.

Figure 2 shows the monthly mean variations in the intensity of $E_s$ layers represented by the S4max index from COSMIC, with superposed ion drifts from the model. The observations of $E_s$ layers from satellites suggest that the maximum $E_s$ latitude has an interhemispheric migration to the summer hemisphere. These $E_s$ seasonal variations have been known for over 50 years (Smith, 1968), but the cause is unknown (Whitehead, 1989; Yu et al., 2019b). The plot also shows a North-South asymmetry (Northern summer peak is stronger than Southern summer peak). The green solid lines are the ratios of column densities of Es

intensities at different latitudes to the global mean column density. Vertical wind shear alone does not account for the variations in relative column densities in $E_s$ layers. The column density of Es layers should be a constant for each month if the $E_s$ seasonal dependence was just a result of a seasonal variation in vertical winds that favor ion convergences in different areas. The peak of the relative column densities shows an interhemisphere transport. This seasonal dependence is consistent with interhemispheric transport of long-lived metallic ions by the winter-to-summer lower thermospheric meridional circulation.

The yellow arrows between 100 and 120 km show the predicted meridional and vertical velocity of ion drift from Eq. (4) & (5), driven by the atmospheric meridional circulations. Above 100 km, metallic ions have a lifetime of at least several days in the thermosphere (Plane et al., 2015). At altitudes of 90–100 km, the lifetime of ions rapidly decrease to time-scales from hours to minutes, because the ions are able to form molecular ions (principally metal oxide ions through reaction with $O_3$). Therefore the predicted velocity of metallic ions is calculated only above 100 km.

It is found that the observed $E_s$ seasonal variation correlates strongly with the ion meridional transport caused by the winter-to-summer lower thermospheric circulation. From February to April, the $E_s$ moves from both hemispheres toward the equator with a large equatorward ion drift speed over 10 m/s. Then $E_s$ moves northward from May to July, and reaches a strong summer maximum, in accordance with the windshear in the lower thermospheric circulation that predicts the winter-to-summer interhemispheric transport of ions. From August to January, $E_s$ migrates toward the equator with the equatorward metallic ion

velocity, followed by a subsequent summer peak in the Southern Hemisphere with a southward ion velocity. Further evidence for a link between the seasonal variation and ion meridional transport is a distinctive northern high-latitude depression of $E_s$, which is not well explained by the vertical windshear effects in previous experimental and theoretical studies (Arras et al., 2008; Chu et al., 2014; Yu et al., 2019b). The dissipative region of $E_s$ during winter solstice agrees with the location at which the mean meridional wind splits between 60°N and 70°N latitude (Figure 1c). According to multi-satellite observations and

simulation results, the seasonal variation in $E_s$ could be highly dependent on the meridional transport of metalic ions, which is inseparable from the ion dynamics of $E_s$ formation with windshear effects.

In addition to neutral winds, the electric field also plays important roles in the transport of metallic ions (Nygren et al., 1984; Bristow and Watkins, 1991; Bedey and Watkins, 2001; Huba et al., 2019; Kirkwood and Nilsson, 2000). A global atmospheric model of meteoric iron (WACCM-Fe) (Feng et al., 2013) reproduces the global seasonal variations of $Fe^+$ density over the

altitude range of Es layers. The $\boldsymbol{E} \times \boldsymbol{B}$ drift, Coulomb-force-induced ion drift (Cai et al., 2019) associated with the electric





field are derived from standard WACMM X (Liu et al., 2018) and the seasonal variation of global distribution of $Fe^+$ is from WACCM Fe (Feng et al., 2013). Figure 3 shows the monthly mean $Fe^+$ flux $\Phi_{Fe^+}$, calculated offline from WACMM X and WACCM Fe. The corresponding latitude-height plots and maps of $Fe^+$ flux due to $\boldsymbol{E} \times \boldsymbol{B}$ drift, Coulomb-force-induced ion drift and neutral winds in January and July are shown in Figures 3a–3b and Figures 3c–3d. The colour scale denotes the

meridional $Fe^+$ flux and extends from -1 (southward) to +1 (northward). This clearly shows a convergence of $Fe^+$ flux between 105 and 115 km at 25°S–45°S in winter and 25°N–45°N in summer. The horizontal red lines in Figures 3a and 3c represent the altitude of 106 km at which the peak density of $E_s$ layers reaches its maximum value and the lower thermospheric meridional circulation dominates. Figures 3b and 3d show the $Fe^+$ flux at this height as a function of latitude and longitude. The $Fe^+$ meridional flux is mostly southward in winter (typically $-3.3 \times 10^7 - -2.3 \times 10^9$ m$^{-2}$s$^{-1}$) and northward in summer (typically

$1.6 \times 10^7 – 4.9 \times 10^9$ m$^{-2}$s$^{-1}$).

Although a variety of ground-based measurements have been performed, most studies interpret observations of earth metallic species in vertical and small-scale transport processes (Macleod, 1966; Plane, 2003; Davis and Johnson, 2005; Johnson and Davis, 2006; Davis and Lo, 2008; Haldoupis, 2012; Chu et al., 2014; Yu et al., 2015; Cai et al., 2017; Yu et al., 2017, 2019a). It is especially important to further monitor the ion-drift motions generated by meridional flows. Signatures in $E_s$ seasonal varia-

tions by ion interhemispheric transport were investigated using the two meridional chains of low-to-middle-latitude ionosonde stations as shown in Figure 4a. Figures 4b–4c show the long-term time-series of $E_s$ layers at each station. The daily mean intensities of $E_s$ layers, by taking a monthly moving average, are plotted from red to blue colors, corresponding to the low-to-middle latitude stations. In particular, all 10 stations exhibit a strong summer peak, which, in the low-latitude stations, appears earlier than in mid-latitude stations. This trend can be used to quantitatively estimate the ion velocity of winter-to-summer

interhemispheric transport.

In Figure 5, green and red points represent annual summer peaks for 10 stations from two meridional chains. Each point represents a day of year (DOY) of the $E_s$ summer maximum. The mean DOY for summer peaks is shown as a box, with the error bar representing the standard error in this mean. An estimation of ion meridional transport was made from the variations in peak time with latitude. Linear fits are presented using measurements from the Greenwich meridional chain (green line),

120°E longitudinal chain (red line), and a total of 10 stations (blue line). A linear correlation is found in the Greenwich meridional chain (correlation coefficient: r=0.38). The p-value is 0.53 calculated for testing the hypothesis of no correlation. The slope is 8.06±0.15 °/day (10.35±0.19 m/s), indicating winter-to-summer ion transport. This value is slightly larger than the predicted meridional velocity in Figure 2; -1.08–7.45 m/s between 10°–60°N in summer. The slope from the 120°E longitudinal chain is 0.99±0.29 °/day∼1.27±0.37 m/s (r=0.99, p<0.01), which is within the predicted range of ion meridional velocity.

This velocity is lower than that from the Greenwich meridional chain, and this is thought to be due to the differences in the magnitude of meridional flows between different solar cycles. The blue line represents the fit for all 10 stations (r=0.67, p=0.04). The estimated ion meridional velocity is 3.16±0.13 °/day (4.06±0.17 m/s). In modeling studies, the dynamics of the lower thermospheric meridional circulation is much less understood than the well-known mesospheric meridional circulation (Liu, 2007; Qian et al., 2017; Qian and Yue, 2017). It is difficult to directly measure the mean meridional wind since the lower

thermospheric meridional circulation is considerably smaller than the tidal amplitudes (Wu et al., 2008). The results presented





here show that the atmospheric circulation has a marked influence on the meridional transport and seasonal variations of the metallic ions in $E_s$ layers. There is also some evidence that the magnitude of the meridional flows may vary with longitude.

## 4 Conclusions

The lower thermospheric winter-to-summer meridional circulation is a process that has generally been ignored, although it plays a fundamental role, comparable to the vertical windshear, in the transport of metallic ions throughout the Earth's upper atmosphere and ionosphere. A difference should be noted though: the distribution of metallic ions, and the formation of the Es. The former is determined by the transport, as demonstrated here in this study, while the latter is probably directly tied to shear. The large-scale horizontal transport of metallic ions needs to be considered over seasonal-to-interannual timescales in global-scale studies of metallic species, and this provides a mechanism that explains the observed $E_s$ seasonal variations (Whitehead, 1960). In addition to the interhemispheric transport, a comprehensive explanation for $E_s$ production requires the use of a whole atmosphere climate model, including all the necessary chemistry and electrodynamics of metallic species. Global models of meteoric metals in the upper atmosphere have been developed by adding the metal chemistry into WACCM (Marsh et al., 2013; Feng et al., 2013; Wu et al., 2019). The results presented here provide an important dynamical constraint for modeling studies that aim to simulate the variations in composition of the thermosphere/ionosphere on seasonal and interannual timescales.

*Data availability.* The COSMIC S4max data are available from the CDAAC website (https://cdaac-www.cosmic.ucar.edu/cdaac/). The ionosonde data are available from the Data Centre for Meridian Space Weather Monitoring Project (https://data.meridianproject.ac.cn), the Institute of Geology and Geophysics, Chinese Academy of Sciences (http://space.iggcas.ac.cn), and the UKSSDC at the Rutherford Appleton Laboratory (https://www.ukssdc.ac.uk). WACCM is an open-source community model available at the NCAR website (http://www.cesm.ucar.edu). The WACCM simulation data in this paper have been archived and are available online (http://mlt.ustc.edu.cn/data/Publications/2020GL089043).

*Author contributions.* B.Y., X.X. and C.J.S. designed the study and wrote the manuscript. J.W. performed the WACCM model runs. W.F., D.R.M., H.L. and J.M.C.P. contributed to the discussion and explanation of model simulations. X.Y. provided the COSMIC radio occultation data and contributed significantly to the comments on an early version in the manuscript. Y.C. and X.D. contributed to the discussion of the results and the preparation of the manuscript. All authors discussed the results and commented on the manuscript at all stage.

*Competing interests.* The authors declare that they have no conflict of interest.

*Acknowledgements.* We acknowledge the Constellation Observing System for Meteorology, Ionosphere, and Climate (COSMIC) Data Analysis and Archive Center (CDAAC) for providing COSMIC radio occultation data, the ionospheric data from the UK Solar System Data Centre





(UKSSDC) at the Rutherford Appleton Laboratory, and the Chinese Meridian Project, the Solar-Terrestrial Environment Research Network, Data Center for Geophysics, Data Sharing Infrastructure of Earth System Science. The authors would like to thank the National Science & Technology Infrastructure of China. The numerical calculations in this paper have been done on the supercomputing system in the Su-percomputing Center of University of Science and Technology of China. This work is supported by the B-type Strategic Priority Program

5      of CAS Grant No. XDB41000000, the National Natural Science Foundation of China (41774158,41974174,41831071,41704148), Anhui Provincial Natural Science Foundation (1908085QD155) and the Fundamental Research Fund for the Central Universities. W.F. and J.M.C.P thank the support from Natural Environment Research Council grant NE/P001815/1. B.Y. would like to acknowledge the Royal Society for the Newton International Fellowship.



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

**Figure 1.** Seasonal dependence of $E_s$ layers and the residual meridional circulations in winter and summer. Global distributions of the mean $E_s$ layer intensity represented by the S4max index from COSMIC are shown in (a) and (b), for the winter (December, January, February) and summer (June, July, August) between 2006 and 2011, detected from COSMIC multi-satellites with a resolution of a $1° \times 1°$. The red and green curves represent the geomagnetic latitude contours of $60°$, $70°$, and $80°$ in the Northern and Southern Hemispheres, and the yellow curve represents the geomagnetic equator. The mean residual circulations are calculated for winter (c) and summer (d) using the SD-WACCM4 (2006–2011). There are three circulation cells: the mesospheric circulation, lower thermospheric circulation and thermospheric circulation. Below 95 km, the summer-to-winter mesospheric circulation is driven by gravity-wave forcing. Between 95 km and 115 km, the winter-to-summer lower thermospheric circulation is driven by gravity-wave forcing that is in the opposite direction to that which drives the mesospheric circulation. Above 115 km, the solar-driven thermospheric circulation is a summer-to-winter circulation.

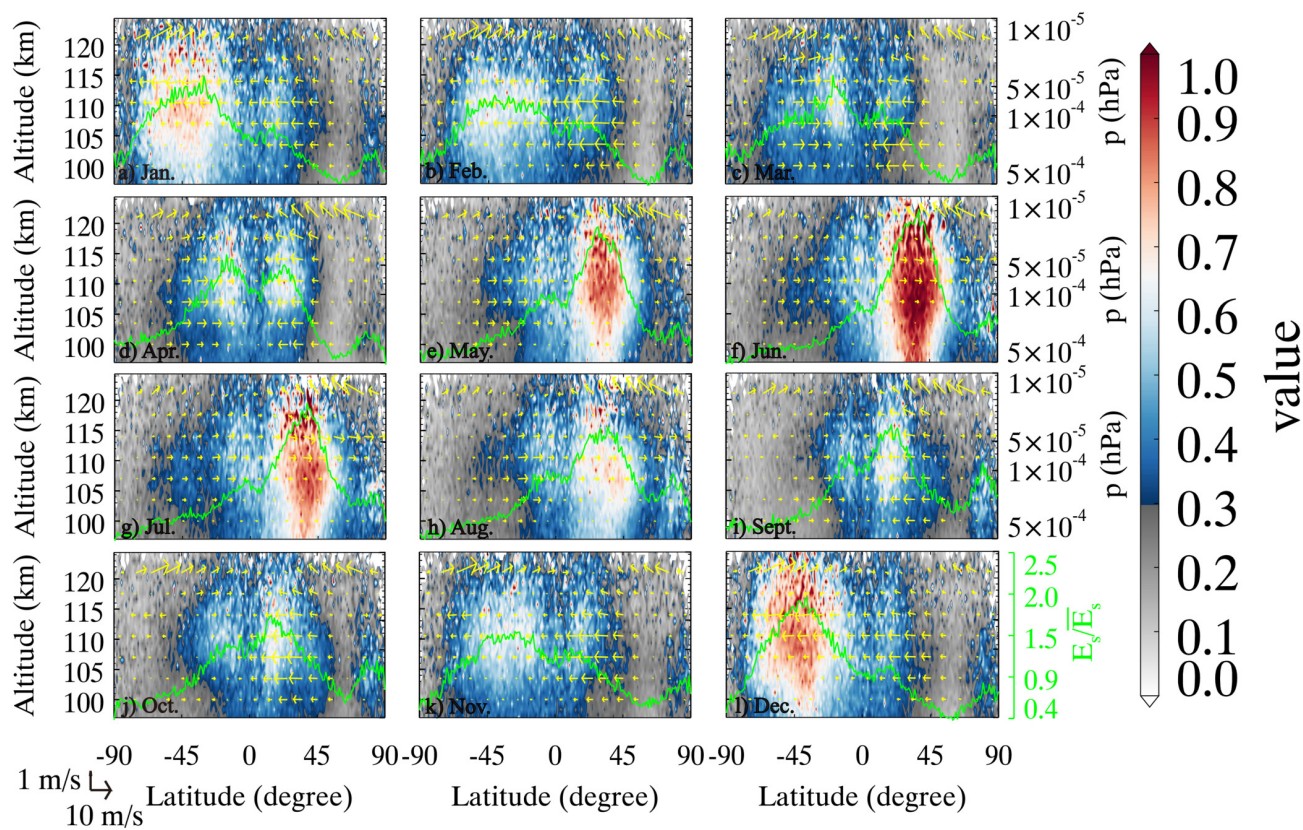

**Figure 2.** Monthly variation in the $E_s$ layer intensity represented by the S4max index from COSMIC, with superposed ion drifts from the model caused by residual meridional circulations. The monthly mean variations in $E_s$ layers are shown for different altitudes and latitudes from 2006 to 2011. The green solid lines are the ratios of column densities at different latitudes to the global mean column density. The yellow arrows between altitudes of 100–120 km show the predicted vertical and meridional velocity of ion drift (vectors with scale $1\text{ms}^{-1}/10\text{ms}^{-1}$), driven by the atmospheric meridional circulations from the SD-WACCM4.

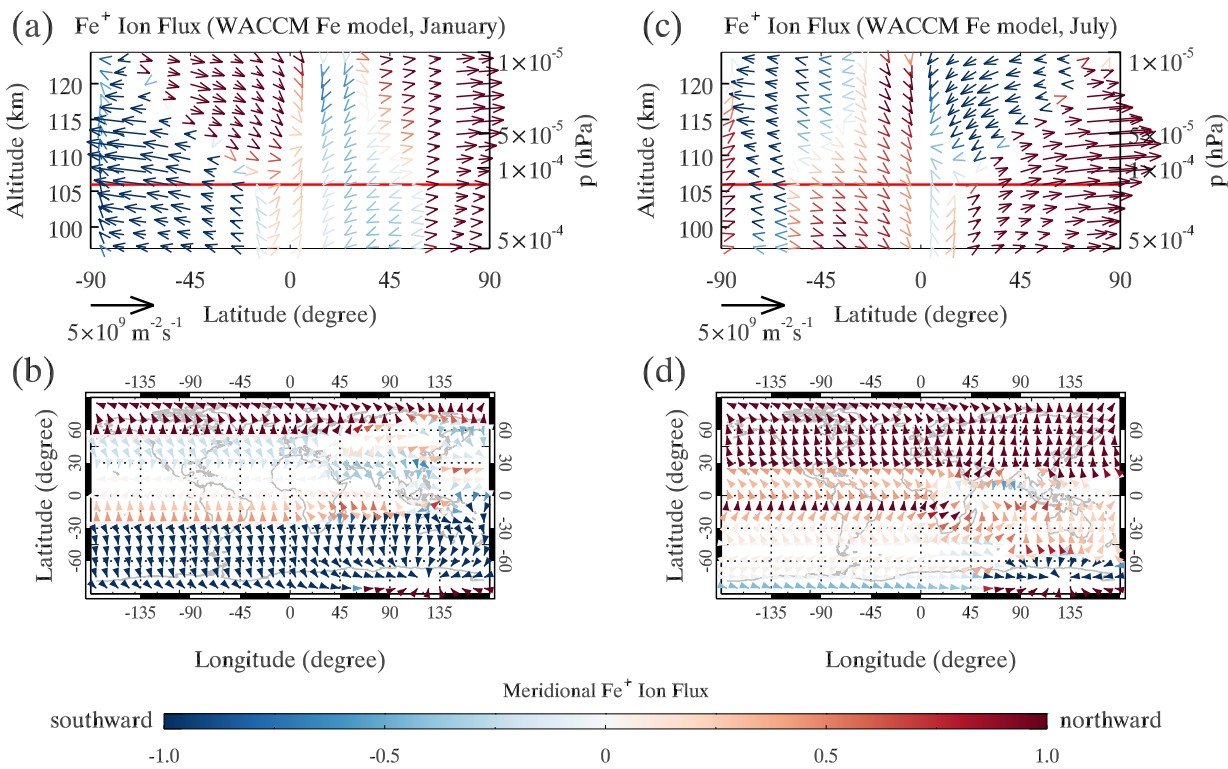

**Figure 3.** (a) Zonal mean latitude-height plot of $Fe^+$ flux $\Phi_{Fe+}$ ($m^{-2}s^{-1}$) calculated offline from WACMM X and WACCM Fe in January, adding $\boldsymbol{E} \times \boldsymbol{B}$ drift, Coulomb-force-induced ion drift and neutral winds effects. The colour scale denotes the meridional $Fe^+$ flux and extends from -1 (southward) to +1 (northward). Also plotted are ion flux vectors which combine the meridional $Fe^+$ flux with the vertical $Fe^+$ flux. The horizontal red line represents the altitude of 106 km at which the peak density of $E_s$ layer reaches its maximum value. (b) map of $Fe^+$ flux at an altitude of 106 km from the model in January. (c) and (d) Zonal mean latitude-height plot and map of $Fe^+$ flux from the model in July.

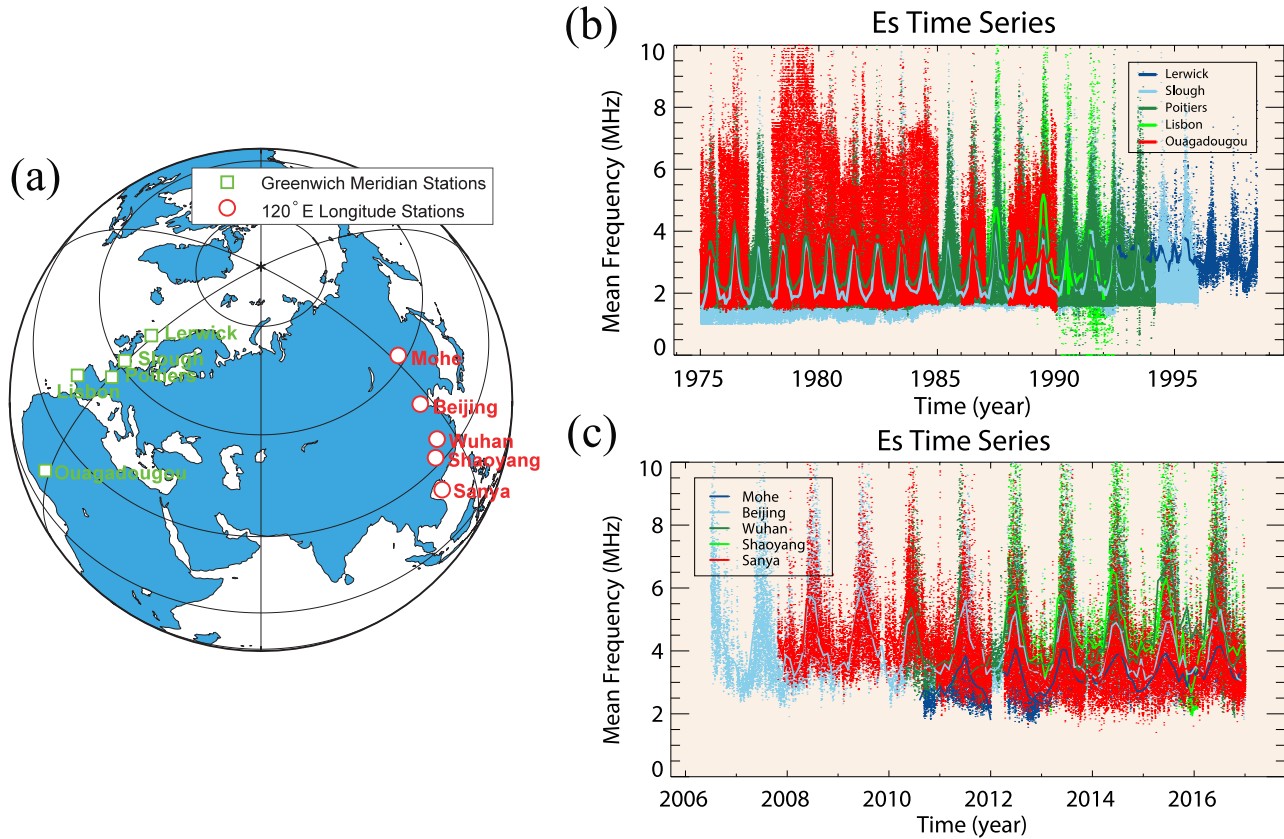

**Figure 4.** (a) Ground-based measurements of $E_s$ from two meridional chains of low-to-middle latitude ionosondes. There are 10 ionosonde stations in these two meridional chains, roughly along the Greenwich meridian (b) and 120°E longitude (c). These ionospheric observations started in 1975. A total of 35-year dataset is analysed to try to identify the signature of the large-scale horizontal transport of metallic ions in $E_s$, induced by meridional circulations.

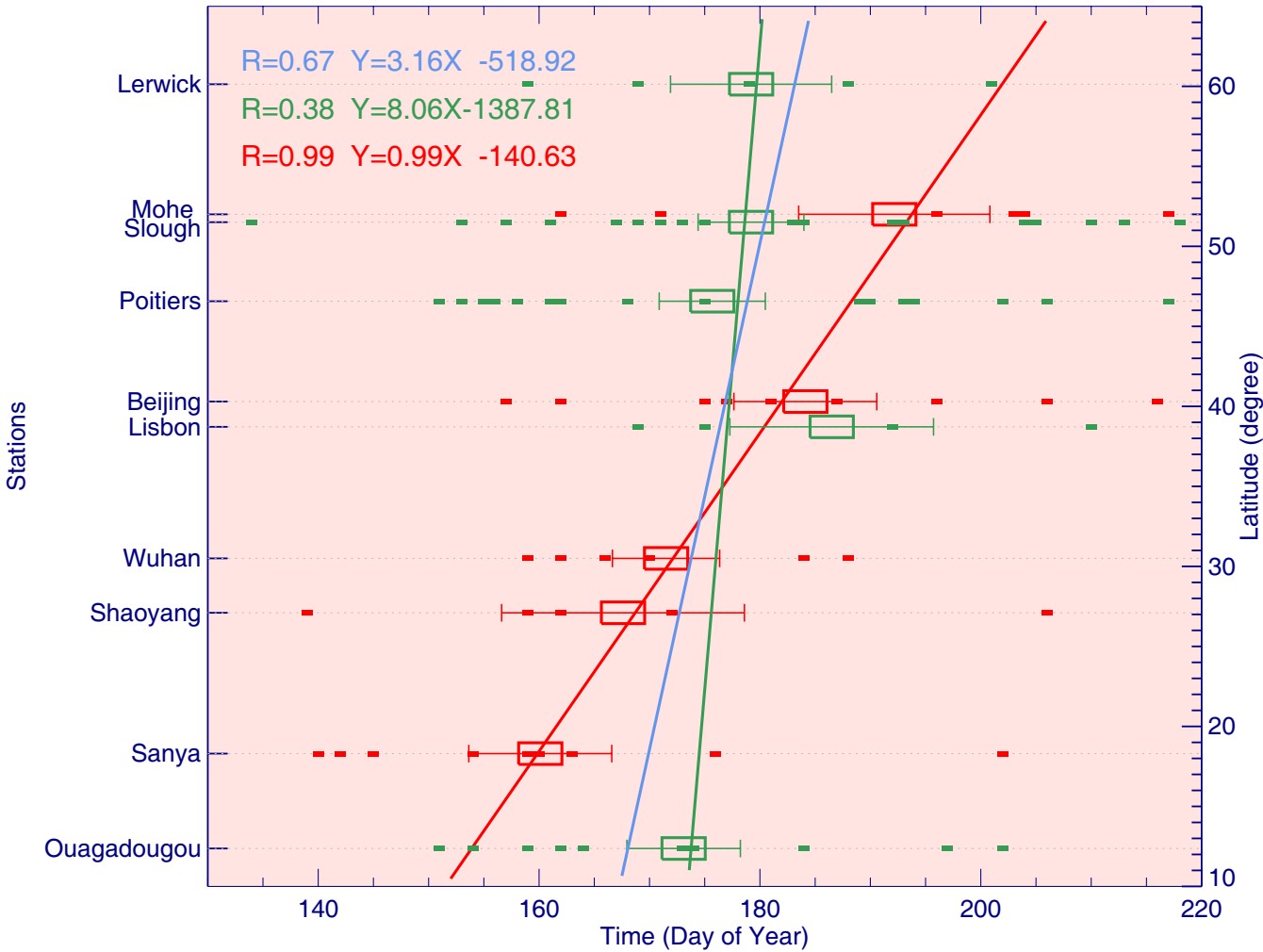

**Figure 5.** Latitudes of stations versus the mean day of year (DOY) time of the annual summer peaks for 10 stations from two meridional chains during 1975–2016. The ionosonde stations cover from 10–60°N latitudes. Green and red points are the annual summer peaks, and each point represents a DOY of a $E_s$ summer peak. The mean DOY time of summer peaks is shown as a box, with the error bar representing the standard error in this mean for each station. The green, red and blue lines show the linear fits between the peak time and latidue, using measurements from the Greenwich meridional chain, 120°E longitudinal chain, and a total of all 10 stations. The linear fit relationship and correclation coefficient are also given, from which an ion meridional velocity in a meridional chain could be quantitatively estimated.