# Peer review of "Interhemispheric transport of metallic ions within ionospheric sporadic E layers by the lower thermospheric meridional circulation"

_Atmospheric Chemistry and Physics, 2020_

## Referee Comment (RC1) · Anonymous Referee #1 · 30 Nov 2020

This paper studies the mechanisms of long lived metallic ions seasonal variations and sporadic E layer formation through COSMIC observations and SD-WACCM simulations. By expanding the vertical shear theory in three-dimension globally, it concludes that the lower thermosphere meridional circulation, usually being ignored, plays deciding role in global metallic ions transport, while the sporadic E layer formation is mainly decided by the wind shear mechanism. Both have to be included in the global model simulation to understand this prominent feature in the E region. This is a very important discovery for the E region dynamics, especially for the nighttime E region variations. On the other hand, the Es intensity is also decided by the peak ion/electron density, so the column abundance alone may not be able to fully characterize the Es seasonal

changes. Although the author mentioned the vertical shear theory, I feel that the role of horizontal wind change over the seasons is not emphasized enough in the paper. The zonal and meridional winds seasonal variations in the E region above $\sim 95$ km make the summer E region dominated by the ion convergence scheme, but divergence scheme in the winter [Yuan et al., 2014]. Minor comments: The choice of 106 km seems to be random, why this altitude is chosen? There are several places in the paper, where the author uses "Fe+ ion". Because "Fe+" itself represents the ion species, no need to add "ion" following it. Please revise. Caption of Figure 1, I suggest the author to delete the first sentence. Caption of Figure 2, please consider to remove "... caused by residual meridional circulations". One question, here. Do you mean the zonal component is not considered in these plots? Also, in these plots, the text of month in red is difficult to read. Please consider to change the color. The scale of "1m/10m" in yellow arrow is hard to read as well. This is scale is neither mentioned in the manuscript nor in the caption, but it is important to know. I suggest the author put a few words somewhere when describing this figure.

---

## Referee Comment (RC2) · Anonymous Referee #2 · 5 Dec 2020

This paper explored the mechanism of seasonal variation of Es by using WACCM wind to calculate the ion drift. The results show that the meridional direction of ion drift is consistent with the inter-hemispheric variation of the Es intensity. Generally, the drift is toward the hemisphere where the Es is high (summer hemisphere). This is a very important finding, and may well explain the seasonal variations of Es. The paper is very well written and the figures are of very high quality.

However, I do find two important issues that need to be addressed:

The ion drift velocity is presumably calculated from the 'residual circulation' from WACCM, which are the zonal mean values. If that is the case, then the ion drift velocity

is not calculated properly because the zonally varying magnetic field and neutral wind are not taken into account. Because of the nonlinear relationship with the magnetic field (D and I) and neutral wind (in Eq.(3)-(5)), ion drift velocities calculated from zonal mean values are not the same as the zonal mean of ion drift velocities calculated from zonally varying D, I and neutral wind field.

Regardless of the above, it will be much clearer to understand the contributions to meridional ion drift (vi) if the three terms in Eq.(4) (from U, V, and W) are shown separately. I suspect that both U and V are the main contributors to vi but their relative importance are different at different D and I values. For example, if U contribution is dominant, then one cannot say that the thermospheric meridional circulation (only related to V and W) is the main transport mechanism.

---

## Author Comment (AC1) · 4 Jan 2021

We would like to thank the reviewers for their valuable comments and suggestions. We have studied all comments carefully and these comments have helped us improve the manuscript. Following the reviewers' comments, we revised the manuscript. Our responses to the reviewers' comments and corresponding changes with page and line numbers in the revised manuscript are both detailed below in blue text. We mark the major changes in red in the track-change manuscript.
* * *
[Figure]

Reviewer #1 comments (RC1): This paper studies the mechanisms of long lived metallic ions seasonal variations and sporadic E layer formation through COSMIC observations and SD-WACCM simulations. By expanding the vertical shear theory in three-dimension globally, it concludes that the lower thermosphere meridional circulation, usually being ignored, plays deciding role in global metallic ions transport, while the sporadic E layer formation is mainly decided by the wind shear mechanism. Both have to be included in the global model simulation to understand this prominent feature in the E region. This is a very important discovery for the E region dynamics, especially for the nighttime E region variations. On the other hand, the Es intensity is also decided by the peak ion/electron density, so the column abundance alone may not be able to fully characterize the Es seasonal changes.

Although the author mentioned the vertical shear theory, I feel that the role of horizontal wind change over the seasons is not emphasized enough in the paper. The zonal and meridional winds seasonal variations in the E region above ∼95 km make the summer E region dominated by the ion convergence scheme, but divergence scheme in the winter [Yuan et al., 2014].

Response: Thank you for your comments. The role of horizontal wind changes in the vertical shear theory over the seasons has been emphasized in the introduction of the revised manuscript. Besides, the individual contributions of zonal, meridional, and vertical winds to meridional and vertical ion drift are shown in new Figure 1. It represents the dominant role of zonal and meridional winds in vertical motion of ions, and the dominant role of meridional wind in meridional motion of ions.

Changes: Please see page 2 lines 5-8. "The most likely explanation for the seasonal dependence of Es layers is the wind shear, by which the seasonal variations of zonal and meridional winds in the E region above 95 km make the summer E region dominated by the vertical convergence of ions, but dominated by the diffusion of ions in winter (Yuan et al., 2014)."

Please see the new Figure 1.

Minor comments: The choice of 106 km seems to be random, why this altitude is chosen?

Response: At altitude of 106 km, the peak density of Es layers reaches its maximum value in Figure 2 and the lower thermospheric meridional circulation dominates.

Changes: Please see page 6 lines 12-14.

There are several places in the paper, where the author uses "Fe+ ion". Because "Fe+" itself represents the ion species, no need to add "ion" following it. Please revise. Response: We have corrected it.

Caption of Figure 1, I suggest the author to delete the first sentence.

Response: Thank you for your comment. The sentence was removed.

Caption of Figure 2, please consider to remove "...caused by residual meridional circulations". One question, here. Do you mean the zonal component is not considered in these plots? Also, in these plots, the text of month in red is difficult to read. Please consider to change the color. The scale of "1m/10m" in yellow arrow is hard to read as well. This is scale is neither mentioned in the manuscript nor in the caption, but it is important to know. I suggest the author put a few words somewhere when describing this figure.

Response: Yes, Figure 2 (Figure 3 in the revised version) shows the contribution of residual meridional circulations to the meridional ion drift. As shown in Figure, the meridional motion of ions is dominantly controlled by the meridional wind. Besides, Figure 4 shows the simulation result considering full wind fields and electric field. Further contributions of zonal, meridional, and vertical winds to meridional ion drift are shown separately in Figure 1.

The text is changed to black color. The scale of ion velocity has been indicated in the

caption of Figure 3. Thank you for your helpful comments.
* * *
Reviewer #2 comments (RC2): This paper explored the mechanism of seasonal variation of Es by using WACCM wind to calculate the ion drift. The results show that the meridional direction of ion drift is consistent with the inter-hemispheric variation of the Es intensity. Generally, the drift is toward the hemisphere where the Es is high (summer hemisphere). This is a very important finding, and may well explain the seasonal variations of Es. The paper is very well written and the figures are of very high quality.

However, I do find two important issues that need to be addressed: The ion drift velocity is presumably calculated from the 'residual circulation' from WACCM, which are the zonal mean values. If that is the case, then the ion drift velocity is not calculated properly because the zonally varying magnetic field and neutral wind are not taken into account. Because of the nonlinear relationship with the magnetic field (D and I) and neutral wind (in Eq.(3)-(5)), ion drift velocities calculated from zonal mean values are not the same as the zonal mean of ion drift velocities calculated from zonally varying D, I and neutral wind field.

Response: Thank you for your comments.

Yes, the result in Figure 3 (formerly Figure 2) is the zonal mean of ion drift velocities induced by residual circulation calculated from zonally varying D and I, but the 'residual circulation' from WACCM is zonal-mean meridional circulation. This result shows that the observed Es seasonal variation correlates strongly with the ion meridional transport caused by the winter-to-summer lower thermospheric circulation.

We also plotted Figure 4 showing a global map of Fe+ flux with the effects of global neutral winds and electric fields varying with D, I. The Fe+ flux is southward in winter and northward in summer.

Regardless of the above, it will be much clearer to understand the contributions to

meridional ion drift (vi) if the three terms in Eq.(4) (from U, V, and W) are shown separately. I suspect that both U and V are the main contributors to vi but their relative importance are different at different D and I values. For example, if U contribution is dominant, then one cannot say that the thermospheric meridional circulation (only related to V and W) is the main transport mechanism.

Response: Thank you for your very helpful comments.

In the revised manuscript, we added one figure (Figure 1) showing the contributions to meridional and vertical ion drift from U, V, and W in January and July separately. The zonal mean of ion drift velocities is calculated from zonally varying D, I and neutral wind field. The ion dynamics become collision-dominated (ri»1) below 125 km altitude (Mathews, 1998). Therefore, the second term with V of Eq. (4) is more important than the other terms, representing the dominant V contribution to the meridional velocity of ions.

Changes: Please see Figure 1 and page 4 lines 16-20.
* * *
[Figure]

[Figure]

Fig. 1. Ion velocity due to zonal (U), meridional (V), and vertical (W) wind in January (a–c) and July (d–f).

---

## Author Response (AR2)

We would like to thank the Editor's comments. The technical corrections have been made accordingly.

The revised manuscript and a track-change version have been uploaded. We mark the changes in red in the track-change manuscript.

We would like to express great appreciation to the Editors and Reviewers for their help in revising our manuscript. Thank you.